# Molecular and morphological survey of Lamiaceae species in converted landscapes in Sumatra

Caitriona Brady Halmschlag[1◉], Carina Carneiro de Melo Moura[1◉]*,
Fabian Brambach[2], Iskandar Z. Siregar[3,4], Oliver Gailing[1,5]*

1 Department of Forest Genetics and Forest Tree Breeding, University of Göttingen, Göttingen, Germany,
2 Biodiversity, Macroecology, and Biogeography, University of Göttingen, Göttingen, Germany,
3 Department of Silviculture, Faculty of Forestry & Environment, IPB University (Bogor Agricultural University), Bogor, Indonesia, 4 Advanced Research Laboratory (ArLab), IPB University (Bogor Agricultural University), Bogor, Indonesia, 5 Centre of Biodiversity and Sustainable Land Use, University of Göttingen, Göttingen, Germany

◉ These authors contributed equally to this work.
* carinamoura@uni-goettingen.de (CCMM); ogailin@gwdg.de (OG)

**Data Availability Statement:** The sequences used in this study were uploaded to NCBI GenBank under the accession numbers: OP481999-OP482063, OP491466-OP491526.

## Abstract

Molecular biodiversity surveys have been increasingly applied in hyperdiverse tropical regions as an efficient tool for rapid species assessment of partially undiscovered fauna and flora. This is done by overcoming shortfalls in knowledge or availability of reproductive structures during the sampling period, which often represents a bottleneck for accurate specimens' identification. DNA sequencing technology is intensifying species discovery, and in combination with morphological identification, has been filling gaps in taxonomic knowledge and facilitating species inventories of tropical ecosystems. This study aimed to apply morphological taxonomy and DNA barcoding to assess the occurrence of Lamiaceae species in converted land-use systems (old-growth forest, jungle rubber, rubber, and oil palm) in Sumatra, Indonesia. In this species inventory, we detected 89 specimens of Lamiaceae from 18 species distributed in seven subfamilies from the Lamiaceae group. One third of the species identified in this study lacked sequences in the reference database for at least one of the markers used (*matK*, *rbcL*, and ITS). The three loci species-tree recovered a total of 12 out of the 18 species as monophyletic lineages and can be employed as a suitable approach for molecular species assignment in Lamiaceae. However, for taxa with a low level of interspecific genetic distance in the barcode regions used in this study, such as *Vitex gamosepala* Griff. and *V. vestita* Wall. ex Walp., or *Callicarpa pentandra* Roxb. and *C. candidans* (Burm. f.) Hochr., the use of traditional taxonomy remains indispensable. A change in species composition and decline in abundance is associated with an increase in land-use intensification at the family level (i.e., Lamiaceae), and this tendency might be constant across other plant families. For this reason, the maintenance of forest genetic resources needs to be considered for sustainable agricultural production, especially in hyperdiverse tropical regions. Additionally, with this change in species composition, accurate species identification throughout molecular assignments will become more important for conservation planning.

**Funding:** All authors were supported by the German Research Foundation (DFG) – project ID 192626868 – SFB 990.

**Competing interests:** The authors have declared that no competing interests exist.

## Introduction

Lamiaceae is the sixth-largest angiosperm family [1] with 236 genera encompassing over 7,000 species [2]. The family consists of both culturally and economically important species, for example used as flavour additives, in cosmetics and as anti-herbivory compounds. This family displays anti-microbial and pollinator attraction functions, which can play an important role in human health and for agricultural purposes [1].

Li et al. [3] and Zhao et al. [4] identified 12 strongly supported primary clades within Lamiaceae. Phylogenetic studies of angiosperms [5] and Lamiales [6, 7] have also placed Lamiaceae within a large clade known as the "core Lamiales" [6], in which Lamiaceae are sister to a well-supported clade made up of Orobanchaceae and other small families such as Mazaceae and Phrymaceae [3]. Lamiaceae can usually be recognized through a combination of traits, which include opposite leaves, bilaterally symmetric flowers with four stamens, and ovaries that consist of two fused carpels that have been divided into a one-seeded chamber [3]. Verbenaceae also display similar traits [3], and it has previously been thought that Lamiaceae had evolved from Verbenaceae-like ancestors [4]. This highlights the importance of molecular phylogenetic studies in distinguishing plant species so that the distinction between similar plant families can be made and any further confusion in classification avoided. Monophyly of clades within Lamiaceae, as well as Lamiaceae's monophyly itself, has been consistently supported through various molecular phylogenetic studies [1, 3, 6–8].

For this study, the samples were obtained from Jambi Province, Sumatra, Indonesia, located in southeast Asia. Sumatra has been reported to be one of the global centers of vascular plant diversity, with 3,000 to 5,000 species per 10,000 km$^2$ [9]. Southeast Asia has higher rates of deforestation when compared to Latin America and Africa [10]. By the year 2100, up to three-quarters of Southeast Asia's original forests and up to 42% of its biodiversity may be lost [11]. In Indonesia, large amounts of forestland have been converted into oil palm plantations over the last few decades, with the emergence of these plantations considered to be a driving force of deforestation [12–15].

Accurately identifying species on a molecular basis, in addition to morphological classification, could become more important with biodiversity loss modelled under certain future scenarios, as in e.g., Sharma et al. [16]. DNA barcoding is a method that can be utilized for species identification and makes use of short DNA regions, often referred to as DNA barcodes [17, 18]. It can be used to distinguish cryptic species [19–21], and in conservation [22, 23]. These DNA barcodes contain variations that can be used to differentiate species [24]. DNA barcoding holds some advantages for species identification over morphological classification, such as its ability to identify organisms regardless of the stage of development [25–27], or a particular gender, as described in e.g., Elsasser et al. [28]. However, one of the limitations of this method is its application to specimen vouchers from herbaria with lower sequence recovery for older specimen material [29]. Additionally, it has been noted in CBOL Plant Working Group et al. [30] that no single locus meets CBOL's data standards and their guidelines for locus selection. Therefore, a combination of loci has been recommended by the CBOL Plant Working Group et al. [30].

The markers used in this study included *matK*, *rbcL*, and ITS. *MatK* is known to have an optimal size, with a high substitution rate and a low transition/transversion ratio [31]. For certain plant species *matK* has been seen to display higher sequence divergence than *rbcL*, however, it can be difficult to amplify by PCR [32]. The advantages of the *rbcL* marker can be seen in its easy amplification and sequencing for many terrestrial plants. It is also an adequate DNA barcoding region for plants at both the family and genus levels [33]. However, *rbcL* has the lowest level of divergence of any of the plastid regions used in Kress et al. [34], but has been

noted to have good, but not excellent discriminatory power [30]. *rbcL*, in combination with various plastid or nuclear loci, can help to make accurate identifications [30, 33, 35]. According to the CBOL Plant Working Group et al. [30], a two-marker combination of the plastid markers *rbcL* and *matK* was recommended as the standard plant barcode [30] with the nuclear ribosomal ITS marker recommended in other studies, such as in Chen et al. [36] and Yao et al. [37]. The ITS marker has been shown to display relatively high universality for angiosperms [38], it displayed the highest overall discriminatory power in comparison with the commonly used chloroplast barcode markers [38]. However, one disadvantage of this marker is that it may also amplify fungal DNA [39].

In this study we conducted a species inventory focusing on the Lamiaceae family in order to generate DNA barcodes as reference sequences for molecular species identification. These were taken from species sampled in four land-use systems (forest, jungle rubber, rubber, and oil palm plots) in Jambi Province, Sumatra. Two DNA chloroplast markers, *matK* and *rbcL*, and the nuclear marker, ITS, were utilized, and thereby the efficiency of these barcode regions for species delimitation in Lamiaceae was evaluated. Due to the high rates of deforestation currently occurring in Indonesia's forests [12–15], the resulting biodiversity loss indicates the urgent need for accurate species identification and reliable reference sequences, in particular for conservation efforts, and thus highlights the significance of this study.

## Materials and methods

### Study site, collection of samples, and taxonomic identification

The Lamiaceae samples were obtained from 32 plots of 50 x 50 m established in four land-use systems (forest, jungle rubber, rubber, and oil palm plantations) located in two landscapes, Bukit Duabelas National Park and the Harapan Rainforest in Jambi province, Sumatra, Indonesia. These sampling sites are made up of the remaining fragments of the tropical lowland rainforests of Sumatra that were once widespread [40] and consist of primary and secondary forests, logged-over forests, including conservation reserve sites in some of the regions, as well as formerly forested areas now converted into agricultural systems.

For species that had previously been identified as Lamiaceae in the field, herbarium specimens were sampled and then mounted to carry out a detailed morphological identification at the Herbarium Borogiense, SEAMEO-BIOTROP Herbarium in Indonesia and at the University of Göttingen in Germany. Fresh leaf tissue was obtained from each sample. This was then dried in silica gel for DNA analysis and transported in sealable bags containing silica gel. In total, 82 leaf samples of 18 species of Lamiaceae were collected and then shipped to the University of Göttingen in Germany for DNA barcoding.

The samples were collected under the approval of the Ministry of Research and Technology and Higher Education of the Republic of Indonesia / National Agency for Research and Innovation, license numbers: 207/SIP/FRP/SM/VI/2012, 25/EXT/SIP/FRP/SM/III/2013, 434/SIP/FRP/E5/Dit.KI/XI/2015, and 42/EXT/SIP/FRP/Dit.KI/VII/2016.

### Extraction, amplification and DNA sequencing

The molecular analysis was employed using one or more samples per species. The DNeasy 96 Plant Kit (Qiagen GmbH, Hilden, Germany) was used for this study, and the DNA was extracted by following the protocol provided by the manufacturer. 20 mg of the dried and healthy leaf tissue of each sample was utilized. These DNA samples were then used for the next steps of the experimental procedure.

The PCR master mix was prepared with a total of 15 μl of the reaction mixture, made up of 1.5 μl PCR-Buffer, 1.5 μL $MgCl_2$ (25mM), 1.0 μl dNTPs (2.5 mM of each dNTPs), 1 μL of

forward primer (5 pM/μL), 1 μL of reverse primer (5 pM/μL), 0.2 μl (5 U/μL) Hot FIREPol Taq (Solis BioDyne, Estonia) and 6.8 μL $H_2O$ with 1μL of the respective DNA sample. The reaction cycle was carried out using a program of 95°C for 15 minutes, 94°C for one minute, 50°C for one minute, 72°C for one minute (go to step two and repeat 34 times) in a Biometra TProfessional thermocycler (Jena, Germany). The samples were then electrophoretically separated on an 1% agarose gel for further quality control. The PCR products were cleaned up prior to sequencing using GENECLEAN Kit (MP Biomedicals). The PCR products were then sequenced using the Big DyeTM Terminator v1.1 Cycle Sequencing Kit (Applied Biosystems).

## Sequence data analysis

Codon Code Aligner software was used to visualise both the forward and reverse sequences for each sample (https://www.codoncode.com/aligner). The sequences were manually checked and then trimmed and edited, where necessary, by checking each electropherogram to examine the quality of the sample at that site. For each sample, consensus sequences were generated, and these were then used for multiple sequence alignments in MEGA7 [41]. BLAST searches were carried out for each consensus sequence generated to find the best matches in the National Centre for Biotechnology Information (NCBI) GenBank [42] and in the Barcode of Life Data system (BOLD) databases [43]. The sequences with a confirmed match were added to the multiple sequence alignments. Representatives of each of the available Lamiaceae tribes [4], as well as representatives of each species were also added. In the case that there was no sequence at the species level present, the corresponding sequences at genus level were added. These compiled sequences were then aligned with MEGA7 [41]. Variable sites, parsimony-informative sites, and the G-C content were calculated following the complete deletion of any missing/gap sites from all sequences [44]. Low-quality sequences and those not assigned to the Lamiaceae family were not used for further analyses or alignments.

The similarity between the obtained sequences and the sequences available in the NCBI Genbank [42] and Barcode of Life Data Systems (BOLD) databases [43] was verified using the BLAST algorithm. Morphological-based identification and molecular assignment were compared to evaluate the efficiency of DNA barcodes through the utilization of reference sequences available in the GenBank and BOLD databases. The percentage of correspondence between morphological and molecular identification results was subdivided into four taxonomic ranks: that of family, tribe, genus, and species.

## Genetic distances and species-tree inference

The calculation of inter- and intraspecific genetic distances was carried out with characterization of Tamura2-Parameter + Gamma in MEGA7 [41] for each single marker (*matK*, *rbcL*, and ITS). The T92 + G model was estimated as the best substitution model for *matK*, *rbcL*, and ITS using MEGA7 [41], with the differences in the transitional and transversional changes and the biased content of G + C [45] taken into account, whereas the speed of evolution between them was modeled by the distribution of gamma (G) [46].

The sequences of each morphologically defined species were grouped to calculate inter- and intraspecific genetic distances. Testing the discriminatory power of each marker was carried out based on the intra-and interspecific divergences of all sequences following previous studies by Wati et al. [24] and Chen et al. [36]. When the minimum value of the interspecific divergence was higher than the maximum level of intraspecific divergence, the barcode was considered informative [47]. The significance of the discriminatory power of each marker was investigated through an unpaired two-samples Wilcoxon test. The Kruskal-Wallis test was

utilised to estimate the significance of the mean differences in intra- and interspecific divergences between the markers ($\alpha$ = 0.05).

The individual *matK*, *rbcL*, and ITS trees were generated using the Maximum Likelihood phylogenetic approach in MEGA7 [41]. The parameters set for the tree in MEGA7 included the bootstrap method with 1000 bootstrap replications. The substitutions type used was 'Nucleotide' and the model/method used was a general time reversible model. The rates among sites selected were Gamma distributed with invariant sites (G+I) with 4 discrete gamma categories. All sites were used. For the heuristic ML method, the nearest-neighbour interchange was utilized. A very strong branch swap filter was applied, as well as one thread. Alignments were concatenated using BioEdit version 7.0.5.3 [48] to estimate a dual-loci tree with the chloroplast markers and a three-loci species tree. Maximum likelihood trees using the concatenated alignments were estimated as previously mentioned. The Interactive Tree of Life (iTOL) [49] was used to annotate final trees.

## Results

### Molecular and morphological taxonomic inventory of Lamiaceae

A total of 18 species belonging to seven subfamilies of Lamiaceae were identified via DNA barcoding and morphological identification: Nepetoideae (N = 1), Callicarpoideae (N = 2), Symphorematoideae (N = 1), Viticoideae (N = 6), Peronematoideae (N = 1), Lamioideae (N = 1), and Ajugoideae (N = 6) (Figs 1 and 2). The highest number of Lamiaceae species was observed in forest (N = 10) and jungle rubber (N = 10). In contrast, a sharp decline in species number was observed in land-use systems with a higher level of agricultural intensification, particularly oil palm plantations (N = 3) (Fig 1A). Interestingly, high interspecific genetic distance was observed in all land-use types, however in forest, the interspecific genetic variation was distributed across a large range, species closely to distantly related were detected in forest plots,

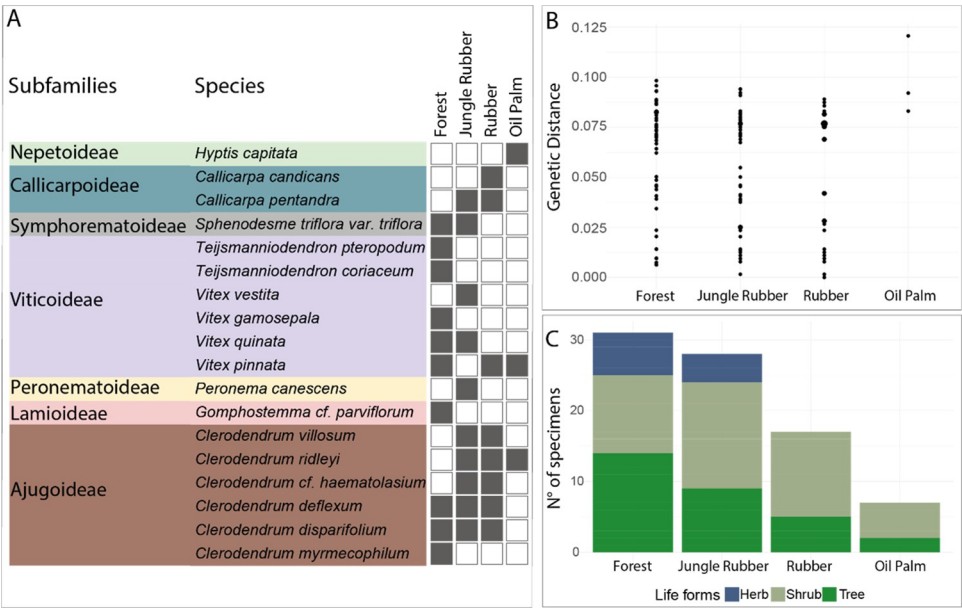

**Fig 1. Overview of molecular and morphological inventory of Lamiaceae. A**- Surveyed Lamiaceae species and their respective occurrence in the four land-use types (forest, jungle rubber, rubber, and oil palm plantations). **B**- Interspecific genetic distance based on *matK* among Lamiaceae taxa encountered in the four land-use types. **C**- Lifeform distribution of Lamiaceae specimens sampled in the four land-use types.

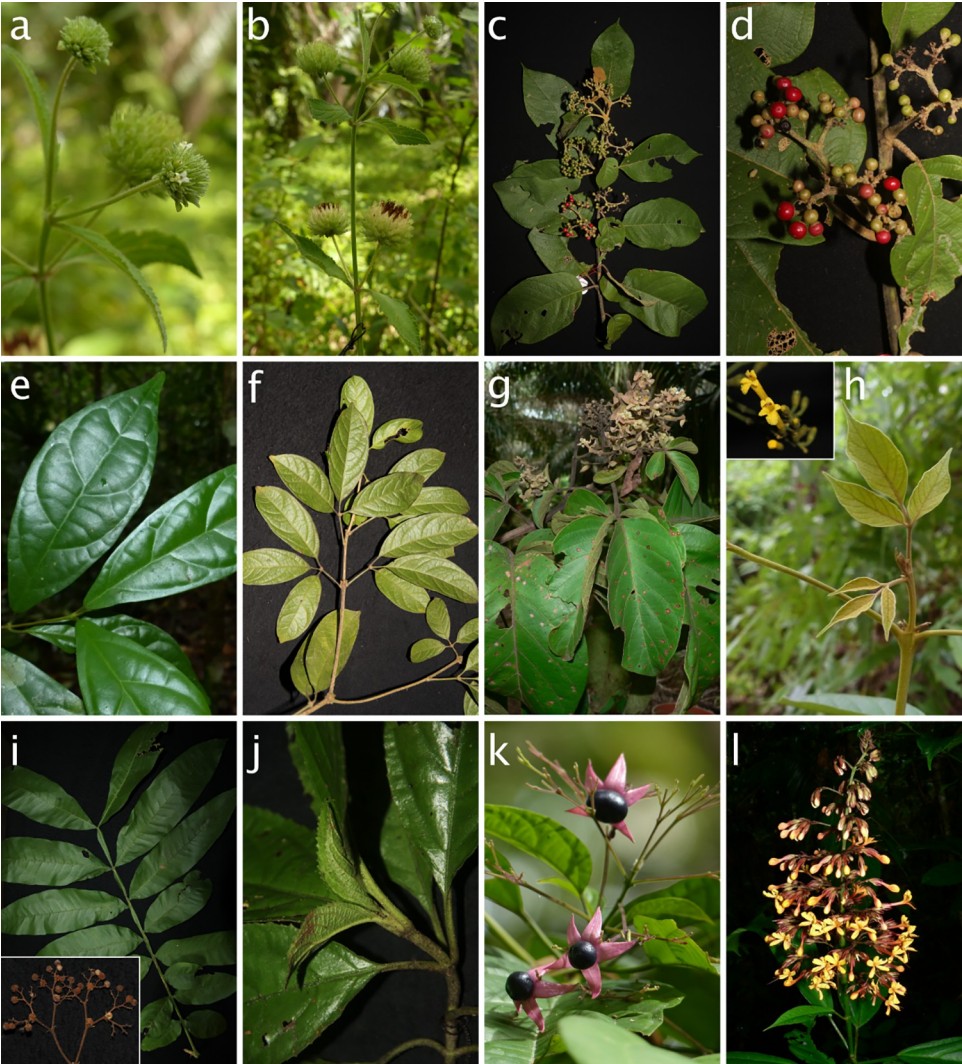

**Fig 2. Morphological variation of Lamiaceae species sampled for this study.** a, b *Hyptis capitata* Jacq. (Nepetoideae), c, d *Callicarpa pentandra* Roxb. (Callicarpoideae), e *Sphenodesme triflora* Wight var. *triflora* (Symphorematoideae), f *Teijsmanniodendron coriaceum* (C.B.Clarke) Kosterm. (Viticoideae), g *Vitex quinata* (Lour.) F.N.Williams (Viticoideae), h *Vitex vestita* Wall. ex Walp. (Viticoideae), i *Peronema canescens* Jack (Peronematoideae), j *Gomphostemma* cf. *parviflorum* Wall. ex Benth. (Lamioideae), k *Clerodendrum disparifolium* Blume (Ajugoideae), l *Clerodendrum myrmecophilum* Ridl. (Ajugoideae).

which reflects the occurrence of species with several degrees of phylogenetic relationships. In contrast, only three species belonging to distinct Lamiaceae subfamilies were found in oil palm plots. One of the three taxa detected in oil palm, *Hyptis capitata* Jacq., is an introduced species in Southeast Asia (Figs 1B and 3). The two more natural systems hold similar species composition, although the number of specimens observed in forest plots (N = 33) was slightly higher than in jungle rubber (N = 28), while rubber plots (N = 20) and oil palm (N = 8) presented lower specimens' numbers and less diversity in life forms (Fig 1A–1C).

The proportion of monophyletic clades and node support increased considerably by combining the datasets as observed in the two-loci and three-loci based species trees (Fig 4 and S1–S4 Figs). Both two-loci and three-loci trees retrieved a similar number of monophyletic clades with 12 resolved species. *Clerodendrum disparifolium* Blume and *C. myrmecophilum*, as well as

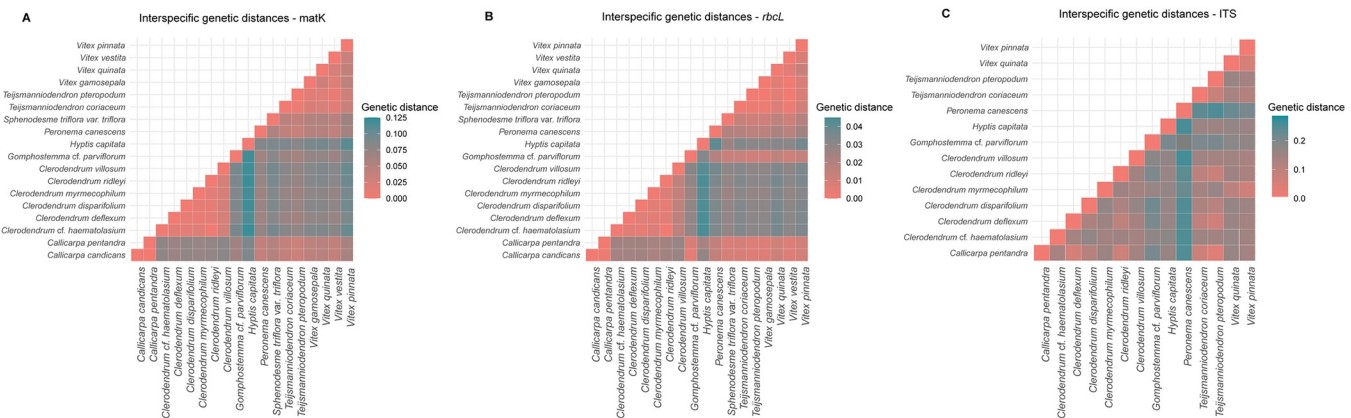

**Fig 3. Heat maps of interspecific genetic distances between Lamiaceae species sampled in Sumatra, Indonesia.** A- Interspecific genetic distances estimated using *matK*, B—*rbcL* and C- ITS regions. Scale bar is adjusted to the maximum genetic distance value of each marker.

*Vitex gamosepala* and *V. vestita*, and *Callicarpa candicans* and *C. pentandra* were found as paraphyletic by the used dataset (Fig 4 and S4 Fig). Juvenile specimens of *Clerodendrum* (KR0625, KR0626, and KR0627) identified as *C. disparifolium* via conventional morphological identification, clustered within *C. myrmecophilum* by DNA barcoding using phylogenetic assignments (Fig 4). After reverification of the specimens due to potential misidentification, the specimens' IDs were corrected to *C. myrmecophilum*. This highlights how DNA barcoding can improve species identification, especially for juvenile plant specimens. The dataset employed in this study is not robust enough to derive conclusions about the phylogenetic relationship of the Lamiaceae family, and the species tree was employed to highlight the applicability of phylogenetic approaches to molecular species assignment. The efficiency of the phylogenetic approaches for taxonomic assignment might vary according to the level of interspecific genetic distance of the barcode markers for closely related organisms.

**Characteristics of DNA barcoding markers.** A total of 73 sequences were successfully obtained for the *matK* marker, 72 sequences for the *rbcL* marker, and 34 sequences for the ITS marker. The ITS dataset displayed the highest proportion of variability (52%), followed by *matK* (39%) and lastly the *rbcL* dataset (18%). The parsimony informative sites showed the same pattern, with ITS having the highest proportion (34%), followed by *matK* (23%), and lastly by *rbcL* (10%), with alignment lengths of 838 base pairs (ITS), 755 base pairs (*matK*) and finally 589 base pairs (*rbcL*) (S1 Table). A total of one-third of the species identified in this study lacked sequences in the reference database for at least one of the markers analysed (*matK*, *rbcL*, and ITS) (S2 Table).

The discriminatory power of the markers *matK* and *rbcL* was effective, with the percentage of identifications at the species level based on the NCBI GenBank BLAST database of 30.8% for *matK*, 22.5% for ITS, and 15.8% for *rbcL*. The percentage of identifications at the species level based on the BOLD BLAST database at the species level was 19.23% for *matK* and 11.53% for *rbcL*. No data for ITS is available in the BOLD database. At the genus level in NCBI, *matK* assigned 89.7% of samples to the correct genus. For *rbcL*, the numbers were similar, with 84.6% of samples assigned to the correct genus in NCBI. For the ITS marker, the numbers were lower, with only 53.75% of samples assigned to the correct genus in NCBI. In the BOLD database, the figures were slightly lower at the genus level, with *rbcL* assigning 79.5% of samples to the correct genus, and *matK* having a slightly lower value than that observed in NCBI, with 76.9% of samples assigned to the correct genus. The markers *matK* and *rbcL* were able to distinguish a large number of taxa at family level (NCBI: *matK* = 93.6%, *rbcL* = 89.7%,

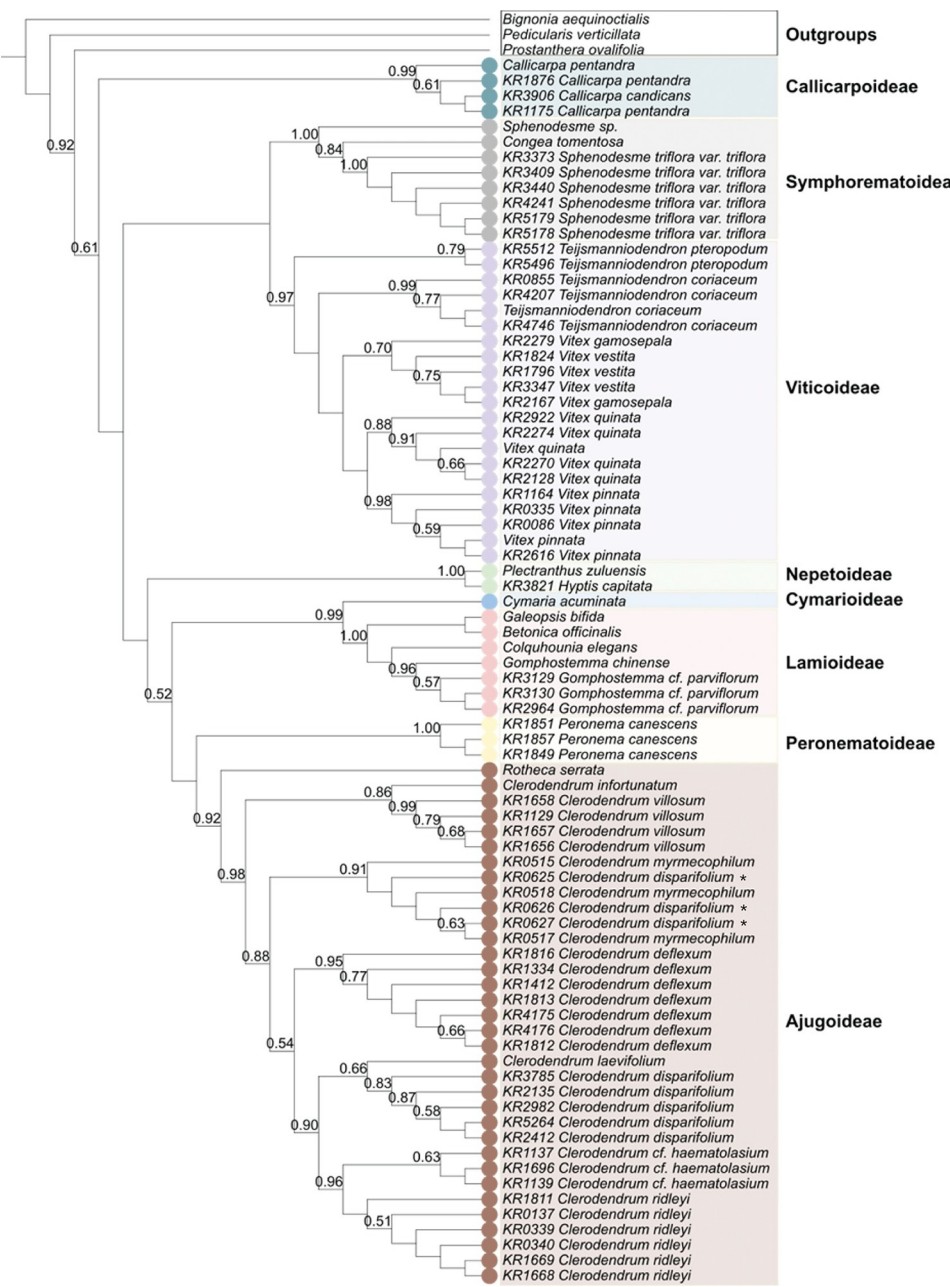

**Fig 4. Maximum likelihood tree of the concatenated sequences of the *matK*, *rbcL*, and ITS markers.** The tips of the tree display the species IDs, with the samples collected for this study highlighted in bold. The subfamilies of the Lamiaceae family are highlighted in different colours. * Highlights juvenile specimens.

ITS = 53.75%), (BOLD: *matK* = 93.6%, *rbcL* = 89.7%). Reference sequences were available for all genera, except for *Sphenodesme* genus for the ITS marker which displayed no results, even at the genus level. There was a lack of reference sequences available in NCBI GenBank and BOLD databases for the species *Clerodendrum cf. haematolasium* Hallier f., *Clerodendrum myrmecophilum*, *Clerodendrum ridleyi* King & Gamble, *Sphenodesme triflora var. triflora*, and *Vitex gamosepala* for all markers. Other species only displayed a lack of reference sequences for certain markers (S2 Table).

Intraspecific genetic distances were significantly lower than the interspecific genetic distances for all markers (*matK*: Kruskal-Wallis chi-squared = 25.996, df = 1, p-value = 3.421e-07; ITS: Kruskal-Wallis chi-squared = 7.0148, df = 1, p-value = 0.008084; and *rbcL*: Kruskal-Wallis chi-squared = 38.735, df = 1, p-value = 4.853e-10). This non-overlap between the two types of genetic distances (intra- and interspecific) confirms the suitability of these barcode markers for molecular species distinction in Lamiaceae. The highest overall mean interspecific genetic distance was observed for the ITS marker, with an overall mean value of 0.142, followed by *matK* with a mean value of 0.06, and then by *rbcL* with a mean value of 0.023. For mean intraspecific genetic distance values, the lowest value was observed for *rbcL* (0.003), then *matK* (0.0084), and finally, ITS (0.082) (Fig 3).

## Conservation status

The conservation status of the species of the Lamiaceae family assessed in our study reported 10 species listed as Least Concern (LC), while eight species were Data Deficient (DD) to categorise their conservation status. Species that are of least concern in the Lamiaceae family make up 48.5% of the Lamiaceae family, with only 4.9% of the family members being near threatened. However, 18% of the family is classified as vulnerable, 14.5% are considered endangered and 10% are classified as being critically endangered. Almost 4% of the species in the present study cannot be accounted for due to data deficiency, as showcased in S3 Table. Despite of that, none of the species assessed in this study is legally protected based on Ministry Regulation of Environment and Forestry of the Republic of Indonesia N˚ 106/2018. Threats to this plant family come from many different sources, such as agriculture and mining practises [50].

## Discussion

A dataset of Lamiaceae samples from Sumatra was used to explore the utilisation of DNA barcoding for species identification in this relatively poorly sampled tropical region and for groups that can be difficult to distinguish. This is due to phenotypic traits that can differ based on environmental factors and development stages [51]. The results obtained in this study indicate that the applicability of DNA barcoding is dependent on both the marker used and the clades analysed. As can be seen in the results listed above, the success of assigning an accurate species identity relies heavily on the presence and availability of reference sequences. One-third of the species used in this study had no reference sequence available in the NCBI GenBank and the BOLD database. The number of reference sequences available also relied heavily on the marker used. The lack of available reference sequences for some of the species used in this study made species delimitation more difficult as a correct assignment at the species level was not possible.

The NCBI RefSeq database has been established as an important resource for genomic, genetic, and proteomic research and the collection provides explicitly linked genome, transcript, and protein sequence records [52]. The level of reference sequence availability will hopefully increase even more in the future and so improve the accuracy of species identification against reference sequences. Herein, we contributed to the NCBI GenBank reference database with 179 sequences from 18 species belonging to the Lamiaceae family, which will facilitate future molecular plant inventories in Sumatra and other distribution areas of the species.

The availability of sequences for certain Lamiaceae species used in this study highlights the importance of effective species identification and will add to the reference databases. In this study, the level of successful assignments was similar between the NCBI GenBank and the BOLD databases, with some differences visible between assignments at the different taxonomic

levels. A lack of reference sequences for certain species, e.g., *Clerodendrum myrmecophilum*, *C. haematolasium*, *Vitex gamosepala*, and several other species has affected the accuracy of taxonomic assignment, as the correct identification of the sequences against reference sequences was not possible. Certain species appeared to be misidentified when searched against the NCBI GenBank and the BOLD databases. As can be seen in the results section, the success rate of an accurate assignment was heavily dependent on the marker used and the database accessed. This highlights the dependency of an accurate result on both the marker used and the number of available reference sequences. NCBI GenBank had a higher number of hits when compared to the BOLD database. We recommend a combined approach of the two databases for the most accurate result, especially considering ITS' absence from BOLD. The increased addition of sequences would greatly add to the information already available.

An optimal barcode can be resolved by a barcoding gap, which arises when the minimum value of the interspecific divergence is higher than the maximum level of intraspecific divergence [47]. One of the most important issues facing DNA barcoding is that of accuracy. The accuracy depends particularly on the degree of, and the separation between intraspecific variation and interspecific divergence in the chosen marker [47]. The results obtained in this study displayed significant results which indicated that the mean interspecific genetic distance was higher than the mean intraspecific genetic distance. This highlights the ability of each of the three barcode markers to differentiate among species of the Lamiaceae family.

Despite difficulties in the processing of samples for DNA barcoding on site, due to the fact that many ecosystems of conservation importance are remote and not easily accessible, and often have no laboratories on site [53], DNA barcoding, through species delineations and identification, has been suggested to improve the effectiveness of conservation planning in e.g., Francis et al. [54] and can also provide information on phylogenetic diversity, as seen in e.g., Gonzalez et al. [55]. DNA barcoding has also displayed a great potential to provide a more efficient biodiversity assessment through the assessment of richness and turnover determined using DNA barcode variability, as shown in e.g., Smith et al. [56]. DNA barcoding as an exclusive method has been shown to be as effective as traditional field identifications in determining species in poorly known floras [57]. With the number of studies on DNA barcoding increasing over the past few years, one can hope that the practicality and accessibility of this research will improve. Increased funding in biodiversity hotspots, many of which are under threat from climate change [58] could vastly improve species identifications and improve conservation efforts in areas facing threats to their biodiversity.

According to the IUCN conservation list, the species used in this study with information available on them are all on the least concern list [50]. However, forest loss and degradation have resulted in Sumatra's primary forest cover loss amounting to 7.54 million hectares (Mha) from 1990 to 2010 with 2.31 Mha primary forest degraded by the year 2010 [59] which could potentially result in an increased rate of species extinction due to increasing deforestation rates [60]. Moreover, many of the species used in this study were declared as data deficient. This highlights the importance of greater numbers of reference sequences and the correct identification of plant species, so that accurate and future estimations of plant species numbers worldwide can be as precise as possible. The misidentification of Lamiaceae species samples based on morphological identification is indicative of the problems caused using a single identification method. Accurate identification of Lamiaceae will greatly aid the conservation efforts of this plant family.

A turnover in species composition linked to the increase in land-use intensification was observed in our study for Lamiaceae. Effects of land-use change may vary across taxonomic groups, favoring generalist species. In this study, the number of Lamiaceae species detected decreases significantly with the reduction of forest cover. Ecological traits which would

provide data to clarify quantitatively if the taxa are indeed ecologically similar is often incomplete or lacking. Consequently, we recommend for further studies the incorporation of ecological trait data to investigate phylogenetic niche conservatism, as a community constituted by closely related species may occupy in sympatry different ecological niches to reduce resource competition, and therefore, the interpretation of phylogenetic composition without ecological data is often mere speculation [61].

Previous studies at the community level in the same study sites revealed a lack of significant intraspecific genetic distances of plant taxa detected in the four land-use systems, however, slightly higher genetic diversity was observed in less intensified land-use systems at plot level [62]. Still, in this study more intensified systems such as rubber and oil palm plantation plots had only a few Lamiaceae tree specimens detected, while less intensified systems i.e., forest and jungle rubber sustained a larger variety of species and abundances in life forms.

## Conclusion

This study highlights the importance of accurate reference sequences. One-third of the species used in this study had no reference sequences available in either of the databases used. 44% of all species used in this study were classified as Data deficient in IUCN's red list. This lack of data is worrying and indicates, once again, the significance of an increased number of sequences that are available in reference databases. The provision of increased reference samples may aid in the conservation effort of Lamiaceae species in Sumatra. An increased number of reference sequences for this study area can aid species identification, can be used to subside conservation plans, act as a model for other studies, and provide additional reasoning for plans to reduce deforestation. Moreover, this study indicated the effect of land-use intensification on Lamiaceae species, with greater species diversity and abundance observed in less intensified systems, such as forest and jungle rubber, and a reduced number of Lamiaceae species in more intensified systems, i.e., rubber and oil palm plantations. These results highlight the negative impact that land-use change can have on species variety and may provide information that could assist conservation planning.

In this study, species-tree inferences of Lamiaceae provided a good alternative for species assignment using a combination of barcode markers, which could be utilized as an extra tool for tropical species delimitation and therefore assist in species identification.

## Supporting information

**S1 Table. Sequence information and characteristics of the individual markers *matk*, *rbcL* and ITS.**
(DOCX)

**S2 Table. The availability of reference sequences for each of the species used in this study.**
(DOCX)

**S3 Table. Conservation status of each species used in this study.**
(DOCX)

**S1 Fig. Maximum likelihood tree estimated based on the *matK* sequences of Lamiaceae.** The tips of the tree labels display the species IDs. The subfamilies of the Lamiaceae family are highlighted in different colours. * Highlights juvenile specimens.
(TIF)

**S2 Fig. Maximum likelihood tree estimated based on the *rbcL* sequences of Lamiaceae.** The tips of the tree labels display the species IDs. The subfamilies of the Lamiaceae family are

highlighted in different colours. * Highlights juvenile specimens.
(TIF)

**S3 Fig. Maximum likelihood tree of the ITS sequences of Lamiaceae.** The tips of the tree labels display the species IDs. The subfamilies of the Lamiaceae family are highlighted in different colours.
(TIF)

**S4 Fig. Maximum likelihood tree based on the chloroplast loci *matK* and *rbcL* of Lamiaceae.** The tips of the tree labels display the species IDs. The subfamilies of the Lamiaceae family are colour highlighted. * Highlights juvenile specimens.
(TIF)

# Acknowledgments

We would like to sincerely thank Gudrun Diederich for her assistance with the technical support in this study. Furthermore, we would like to thank the Ministry of Research, Technology and Higher Education of the Republic of Indonesia (RISTEKDIKTI)/ National Agency for Research and Innovation (BRIN) for the granting of permission to perform the research in Indonesia. This research is part of the Collaborative Research Centre 990 –EFForTS (Ecological and Socioeconomic Functions of Tropical Lowland Rainforest Transformation Systems, https://www.uni-goettingen.de/efforts). We acknowledge the Open Access Publication Funds of the University of Göttingen.

# Author Contributions

**Conceptualization:** Caitriona Brady Halmschlag, Carina Carneiro de Melo Moura, Iskandar Z. Siregar, Oliver Gailing.

**Data curation:** Caitriona Brady Halmschlag, Carina Carneiro de Melo Moura, Fabian Brambach.

**Formal analysis:** Caitriona Brady Halmschlag, Carina Carneiro de Melo Moura.

**Funding acquisition:** Oliver Gailing.

**Investigation:** Caitriona Brady Halmschlag, Carina Carneiro de Melo Moura, Fabian Brambach, Iskandar Z. Siregar.

**Methodology:** Caitriona Brady Halmschlag, Carina Carneiro de Melo Moura.

**Project administration:** Carina Carneiro de Melo Moura, Iskandar Z. Siregar, Oliver Gailing.

**Resources:** Carina Carneiro de Melo Moura.

**Software:** Caitriona Brady Halmschlag, Carina Carneiro de Melo Moura.

**Supervision:** Carina Carneiro de Melo Moura, Oliver Gailing.

**Validation:** Carina Carneiro de Melo Moura, Fabian Brambach.

**Visualization:** Carina Carneiro de Melo Moura, Fabian Brambach.

**Writing – original draft:** Caitriona Brady Halmschlag, Carina Carneiro de Melo Moura.

**Writing – review & editing:** Caitriona Brady Halmschlag, Carina Carneiro de Melo Moura, Fabian Brambach, Iskandar Z. Siregar, Oliver Gailing.

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
