## [Editor Report · Decision Letter 0]

28 Jul 2022

PONE-D-22-11075Molecular and morphological survey of Lamiaceae species in converted landscapes in SumatraPLOS ONE

Dear Dr. Moura

Thank you for submitting your manuscript to PLOS ONE. After careful consideration, we feel that it has merit but does not fully meet PLOS ONE’s publication criteria as it currently stands. Therefore, we invite you to submit a revised version of the manuscript that addresses the points raised during the review process.

We look forward to receiving your revised manuscript.

Kind regards,

Arun Kumar Jugran, Ph. D.

Academic Editor

PLOS ONE

Journal Requirements:

All authors were supported by the German Research Foundation (DFG) – project ID 192626868 – SFB 990.

acknowledge support by the German Research Foundation (DFG, 

German Research Foundation) – project ID 192626868 – SFB 990.

However, funding information should not appear in the Acknowledgments section or other areas of your manuscript. We will only publish funding information present in the Funding Statement section of the online submission form. 

All authors were supported by the German Research Foundation (DFG) – project ID 192626868 – SFB 990.

Additional Editor Comments:

The manuscript entitled “Molecular and morphological survey of Lamiaceae species in converted landscapes in Sumatra" comprises informative and valuable work. The study is useful in view of harnessing the potential of the species of Lamiaceae in Sumatra and other region of the world. It is also important to resolve taxonomic ambiguities within the member of this family and help to establish phylogenetic relationships among the related species/genera. Manuscript is well written and presented nicely. Data analysis is sufficient to draw conclusion and interpretation of the data is good. Based on observations during reviewing this manuscript, I recommend the manuscripts for minor revision to enhance the scope of the MS among readers. To improve the quality and scope of the paper, the following minor comments and suggestions are required to be addressed by the authors

Abstract should be more clear and appropriate. Rewrite it should have one or two concluding line and recommendation based on this study.

Reduce the size of introduction it’s too lengthy.

How the leaf sample was collected from the field. Zip lock bag with slica gel was used to transport the samples from field to lab or anything else please describe in the relevant section

Replace mix with master mix in method.

Conclusion part also require more precise summary of the results and expected outcomes of the study.

Overall, manuscript is well written pure scientific and provides novel information and hence recommended for acceptance with minor revision.
---

## [Author Response · Author response to Decision Letter 0]

19 Oct 2022

Dear Editor, 

We included in the current version the license details including the full name of the authority (Ministry of Research and Technology and Higher Education of the Republic of Indonesia / National Agency for Research 

and Innovation). 

Accession numbers were also included.

See details below:

Lines 132 to 135:

"The samples were collected under the approval of the Ministry of Research and 

Technology and Higher Education of the Republic of Indonesia / National Agency for Research 

and Innovation, license numbers: 207/SIP/FRP/SM/VI/2012, 25/EXT/SIP/FRP/SM/III/2013, 

434/SIP/FRP/E5/Dit.KI/XI/2015, and 42/EXT/SIP/FRP/Dit.KI/VII/2016." 

Line 418 to 419:

The sequences used in this study were uploaded to NCBI GenBank under the accession 

numbers: OP481999-OP482063, OP491466-OP491526.

Many thanks for your time and consideration.

Sincerely, 

Carina Moura

---

## [Editor Report · Decision Letter 1]

3 Nov 2022

Molecular and morphological survey of Lamiaceae species in converted landscapes in Sumatra

PONE-D-22-11075R1

Dear Dr. Carneiro de Melo Moura,

We’re pleased to inform you that your manuscript has been judged scientifically suitable for publication and will be formally accepted for publication once it meets all outstanding technical requirements.

Kind regards,

Arun Kumar Jugran, Ph. D.

Academic Editor

PLOS ONE

Additional Editor Comments (optional):

The manuscript entitled “Molecular and morphological survey of Lamiaceae species in converted landscapes in Sumatra" is revised properly and address all the comments raised by the reviewer and editor. It contains valuable information, work and will help to resolve taxonomic ambiguities within the member of this family. Point wise clarification provided by the authors is satisfactory and addressed in the proper section of the text. Based on observations during reviewing revised version of this manuscript, I strongly recommend the manuscripts for publication.
---

## [Editor Report · Acceptance letter]

6 Dec 2022

PONE-D-22-11075R1 

Molecular and morphological survey of Lamiaceae species in converted landscapes in Sumatra 

Dear Dr. Carneiro de Melo Moura:

I'm pleased to inform you that your manuscript has been deemed suitable for publication in PLOS ONE. Congratulations! Your manuscript is now with our production department. 

Kind regards, 

on behalf of

Dr. Arun Kumar Jugran 

Academic Editor

PLOS ONE